# Detection of Anti-Vimentin Antibodies in Patients with Sarcoidosis

**DOI:** 10.3390/diagnostics12081939

**Published:** 2022-08-11

**Authors:** Anna Starshinova, Anna Malkova, Ulia Zinchenko, Sergey Lapin, Alexandra Mazing, Dmitry Kudlay, Piotr Yablonskiy, Yehuda Shoenfeld

**Affiliations:** 1Laboratory of the Mosaic of Autoimmunity, Medical Department, St. Petersburg State University, 199034 St. Petersburg, Russia; 2Phthisiopulmonology Department, St. Petersburg Research Institute of Phthisiopulmonology, Ministry of Health of the Russian Federation, 127994 St. Petersburg, Russia; 3Laboratory for the Diagnosis of Autoimmune Diseases of the Scientific and Methodological Center, St. Petersburg State Medical University, 194100 St. Petersburg, Russia; 4Department of Pharmacology, I.M. Sechenov First Moscow State Medical University, 119435 Moscow, Russia; 5Laboratory of Personalized Medicine and Molecular Immunology, NRC Institute of Immunology FMBA of Russia, 115478 Moscow, Russia; 6Zabludowicz Center for Autoimmune Diseases, Sheba Medical Center, Tel Hashomer, Ramat Gan 52621, Israel; 7Sackler Faculty of Medicine, Tel-Aviv University, Tel Aviv 69978, Israel

**Keywords:** sarcoidosis, granulomatous diseases, vimentin, mutated vimentin, autoantibodies, autoimmunity

## Abstract

There is a need to further characterize the antibody response to vimentin in relation to its possible involvement in pathogenicity of sarcoidosis and other lung disorders. Objectives: We investigated serum samples from patients with sarcoidosis, healthy controls and controls with other non-infectious lung diseases., to evaluate levels and frequency of these antibodies. Materials and methods: A retrospective-prospective comparative study was performed in the years 2015–2019. Sera from 93 patients with sarcoidosis, 55 patients with non-infectious lung diseases and 40 healthy subjects was examined for presence of autoantibodies to mutated citrullinated vimentin (anti-MCV). Patients with elevated anti-MCV levels were tested for antibodies to a cyclic citrullinated peptide (anti-CCP) and citrullinated vimentin (anti-Sa). In all cases ELISA assays was used. The results were considered statistically significant at *p*-value less than 0.05. Results of the study: The high concentrations of anti-MCV antibodies were more frequent in patients with sarcoidosis (40.9% of the cases, 38/93), compared to the control groups (23.6% and 25.0% of cases, respectively). In sarcoidosis, clinical symptoms similar to the autoimmune pathology were described. A moderate positive correlation between the anti-MCV and anti-Sa antibodies (r = 0.66) was found in 13 patients with sarcoidosis. There was no significant difference between the levels of the anti-MCV and the anti-CCP in patients with non-infectious lung diseases and the healthy control group. Conclusion: Antibodies to citrullinated cyclic peptides are not significant in the pathogenesis of sarcoidosis and other investigated pulmonary diseases (COPD, granulomatosis with polyangiitis, alveolitis) and based on their low concentration, it can be assumed that citrullination and modification of vimentin is not a key factor in the development of an autoimmune response in patients with sarcoidosis.

## 1. Introduction

Sarcoidosis is a granulomatosis of an unknown etiology, typically presented as a systemic relatively benign disease, characterized by the accumulation of activated T lymphocytes (CD4+) and mononuclear phagocytes with the formation of epithelioid cell granulomas [1,2,3]. The lesions in the lungs are the most common clinical symptom, reported in 90% of the patients, followed by arthritis and lymphadenopathy. In rare cases, the bone tissue, mucous membranes, skin and liver are also involved. The disease can be acute (Löfgren’s syndrome), characterized by symptoms of erythema nodosum, bilateral mediastinal lymphadenopathy, articular syndrome, fever, or having a chronic course with development of fibrotic changes in the lung tissue [3,4].

Sarcoidosis is described worldwide, varying in frequency, prevalence and clinical symptoms between regions and populations. Genetic differences between nationalities and races and difficulties in diagnostics because of the unclear etiology of the disease could be the explanation for these diversities [4,5,6]. Air pollution and other environmental factors, as well as infectious agents (*Propionibacterium acnes*, *Mycobacterium tuberculosis*), are considered as causative factors [4].

Pulmonary tuberculosis is one of the diseases that usually requires differential diagnosis with sarcoidosis. The absence of specific clinical, laboratory and radiological symptoms of sarcoidosis usually requires morphological verification and performing the immunohistochemical methods [3].

In recent years, it has been suggested that sarcoidosis is an autoimmune disease [7,8,9,10,11,12]. This can be confirmed by the detection of different autoantibodies in the blood, association with the HLA genotype, lymphoid infiltration, and the response to an immunosuppressive therapy [13,14,15]. Several autoantibodies were reported in sarcoidosis: anti-mitochondrial antibodies, anti-nuclear antibodies and rheumatoid factor in the blood or in the bronchoalveolar fluid of patients with the disease [16,17,18,19,20,21,22,23].

In the past few years, vimentin, a mesenchymal cytoskeletal cell filament protein, has gained the attention as a potential autoantigen in this disease. Vimentin is a cytoskeletal component that presents in the connective tissue and participates in the intercellular interactions and the functioning of the immune system [24,25,26]. The occurrence of autoantibodies to this protein was observed in rheumatoid arthritis, systemic lupus erythematosus and many other connective tissue diseases [26,27,28].

Enzymatic citrullination of vimentin in vitro leads to the formation of modified citrullinated vimentin (MCV). Antibodies to mutated citrullinated vimentin (anti-MCV) are a member of a large family of anticitrullinated antibodies (filaggrin, anti-perinuclear factor, citrullinated vimentin (or Sa antigen) purified from the placenta). Among them, the most commonly used anticitrullinated antibodies are anticyclic citrullinated peptide antibodies (anti-CCP) and antibodies against MCV [26].

The autoimmune nature of sarcoidosis needs further evaluation, thereby the purpose of this study was to determine the presence of autoantibodies to various vimentin modifications in patients with sarcoidosis and other non-specific lung diseases.

## 2. Materials and Methods

### 2.1. Study Population

A retrospective comparative study based on the evaluation of serum samples collected in the years 2015–2019 at the St. Petersburg Research Institute of Phthisiopulmonology and the City Hospital No. 2 was conducted. The laboratory experiments were performed at the Laboratory for Diagnosis of Autoimmune Diseases at St. Petersburg State Medical University.

In the study 188 patients were recruited, of which 93 participants had histologically verified pulmonary sarcoidosis with the involvement of intrathoracic lymph nodes. The demographic characteristics of the patients with sarcoidosis are presented in the Table 1. Disease control groups were: 55 with non-infectious lung diseases; 25 patients with chronic obstructive pulmonary disease (COPD); 15 patients with granulomatosis with polyangiitis; and 15 with various alveolitis. The additional control group was comprised of 40 healthy volunteers with no chronic diseases, contacts with tuberculosis and changes in laboratory parameters.

All groups of patients were sex and age matched. Exclusion criteria included a period of more than 2 years from the detection of radiographic changes in the lungs, receiving anti-tuberculosis and immunosuppressive therapy, conducting a course of plasmapheresis less than 2 months prior to the date of the inclusion, the presence of HIV infection, syphilis, neoplastic diseases, and decompensated diabetes mellitus.

The study was approved by the Independent Ethical Committee of the St. Petersburg Research Institute of Phthisiopulmonology (protocol No. 34.2 dated 19 January 2017) and the Local Ethical Committee of St. Petersburg State University (protocol No. 01-126 30.06.17). All the participants included in the study signed an informed consent.

### 2.2. Methods

All patients underwent physical examination, multispiral chest computed tomography (MSCT), laboratory blood tests, standard tuberculosis screening tests, T-SPOT.TB test, histological verification of the lung and intrathoracic lymph nodes lesions (using a transbronchial and videothoracoscopic biopsy). The diagnosis of lung sarcoidosis was determined according to the standard criteria of the American Thoracic Society (ATS), the European Respiratory Society (ERS) and the World Association of Sarcoidosis and Other Granulomatous Disorders (WASOG) [29].

### 2.3. Anti-Vimentin Antibodies Determination

The levels of antibodies to mutated citrullinated vimentin (anti-MCV) were measured in the sera of all the participants. Patients positive for anti-MCV antibodies were evaluated for the presence of antibodies to cyclic citrullinated peptide (anti-CCP); citrullinated vimentin (anti-Sa) and non-modified vimentin.

Antibodies to anti-MCV were measured using ELISA (ORGENTEC Diagnostika, Mainz, Germany), anti-CCP and anti-Sa—both with ELISA (EUROIMMUN, Lübeck, Germany). For evaluation of positive autoantibody concentration, the cutoffs proposed by the manufacturers were used. Additionally, anti-non-modified vimentin autoantibodies were assessed by using a sandwich-ELISA (Elabscience Biotechnology Inc., Houston, TX, USA) as recommended by the manufacturer. The antibody results were considered positive if their concentration was above 89.2 RU/mL as determined by receiver operative curve (ROC) analysis (data not shown).

### 2.4. Statistical Analysis

Statistical analysis was performed using GraphPad Prism 6 (Graph Pad Software, San Diego, CA, USA), Statistica 10 (Statsoft, Tulsa, OK, USA) and MedCalc—version 18.2.1 (Ostend, Belgium) values.

The Mann–Whitney U and Fisher’s exact tests were used for non-parametric data. Quantitative data were presented as M ± SD. The degree of association was calculated using confidence intervals, as well as the χ2 test with Yeats correction. To determine the relationship between the values, Spearman correlation analysis was performed. Differences or relationship indicators were considered statistically significant at a *p*-value less than 0.05.

## 3. Results

The levels of anti-MCV and anti-CCP autoantibodies in the sera of the participants are presented in the Table 2.

In 40.9% (38/93) patients with sarcoidosis the levels of anti-MCV were elevated, the difference between these groups was statistically significant (*p* = 0.027). The proportion of patients with elevated levels was higher than for disease controls (20.0%) or healthy controls (7.5%), *p* < 0.0001. The anti-CCP antibodies were detected only in one patient with sarcoidosis and in two patients with other lung diseases and in the control groups. Anti-Sa antibodies were presented in the group of patients with sarcoidosis (*n* = 13) and in the subjects with non-specific lung diseases (*n* = 9). A high concentration of these antibodies was detected in seven patients with sarcoidosis and in two patients with non-infectious lung diseases (Table 3).

A moderate positive relationship (r = 0.66, *p* = 0.017) was found between the anti-MCV and anti-Sa autoantibody levels in the 13 antibody-positive patients with pulmonary sarcoidosis (Figure 1).

Significant difference was found when comparing the anti-MCV antibody concentrations in patients with sarcoidosis and non-infectious lung disease (*p* = 0.0003). A significant difference was also found in patients with tuberculosis and non-infectious lung diseases (*p* < 0.0001). The levels of autoantibodies to non-modified vimentin were significantly higher (*p* = 0.03) in comparison with the control group (Figure 2).

Almost all patients with Löfgren syndrome included in the study had high levels of anti-MCV antibodies, which was statistically more frequent than in patients without Löfgren’s syndrome (Table 4).

A prevalence of non-specific symptoms that can characterize autoimmune pathology (arthritis, dry mouth, sleep disorders, memory disturbance, fever, general weakness, chronic fatigue, etc.) was reported in anti-MCV antibodies positive patients (5.4 vs. 1.3 symptoms (*p* = 0.03)).

We also compared smokers and non-smokers among the patients with sarcoidosis. The influence of previous or present smoking to citrullination was not statistically significant (Figure 3).

## 4. Discussion

The pathogenesis of sarcoidosis is commonly considered mediated by a cellular immune response involving CD4+ T-helper lymphocytes that characteristically compartmentalize in the lungs. However, the proposal of vimentin as an autoantigen in sarcoidosis suggests autoimmune involvement [8,24].

The anti-MCV antibodies were defined in patients with sarcoidosis and tuberculosis and not in the control group, which could imply an autoimmune mechanism in pathogenesis of these diseases [30,31,32,33].

In 2007, Wahlström’s group described possible autoantigens associated with HLA-DR molecules presented by bronchoalveolar fluid cells in patients with sarcoidosis [10]. A pronounced response of T-cells of the bronchoalveolar fluid to vimentin and citrullinated vimentin was also demonstrated in patients with the HLA genotype DR-B1*0301 and acute forms of the disease, which was confirmed in patients with Löfgren’s syndrome in our study [9].

The reason for the elevation of autoantibodies may be molecular mimicry, which is described in chronic infections with a hyperactive immune response. According to the different studies, vimentin is a possible autoantigen that activates both cell and humoral autoimmune response in patients with sarcoidosis [34,35]. These findings are consistent with our data. The heat shock proteins Mtb-HsP60, Mtb-HsP65, and catalase (mKatG) can be considered as the candidate mycobacterial antigens, possibly involved in the cross-reaction [36,37].

The presence of antibodies in patients with tuberculosis and sarcoidosis may reflect the relationship between the pathogenesis of those diseases with the possibility of cross-reactivity between vimentin and *M. tuberculosis* peptides [38,39,40].

The results of this study suggest that citrullination, as well as vimentin modification, are not the main triggers in the initialization of an autoimmune reaction in patients with sarcoidosis. This finding may be crucial in better understanding disease mechanisms and therefore for developing an appropriate treatment of patients with sarcoidosis.

The study has the following limitations: inclusion in the group of patients with Löfgren’s syndrome, in whom the antibody detection rate was higher than in patients without Löfgren’s syndrome; the need to evaluate a larger sample to analyze the effect of smoking on citrullination.

## 5. Conclusions

In the present study we investigated the levels and frequency of autoantibodies to different modifications of vimentin (MCV, Sa and non-modified vimentin) and CCP in serum from patients with sarcoidosis and other lung disorders. The detection of anti-MCV antibodies cannot serve as a diagnostic criterion for these diseases; however, the identification of an autoimmune component (antibodies to vimentin) in the pathogenesis of these diseases may be important for an appropriate treatment of patients. Therefore, appearance of anti-vimentin (anti-MCV, anti-Sa) antibodies may serve as a marker for the administration of immunosuppressive therapy.

For the first time, it was shown that antibodies to citrullinated cyclic peptides are not significant in the pathogenesis of sarcoidosis and other investigated pulmonary diseases (COPD, granulomatosis with polyangiitis, alveolitis). The low concentration of the anti-CCP antibodies and the positive correlation of anti-MCV and anti-Sa antibodies suggest that citrullination and modification of vimentin is not a key factor in the development of an autoimmune response in patients with sarcoidosis.

## Figures and Tables

**Figure 1 diagnostics-12-01939-f001:**
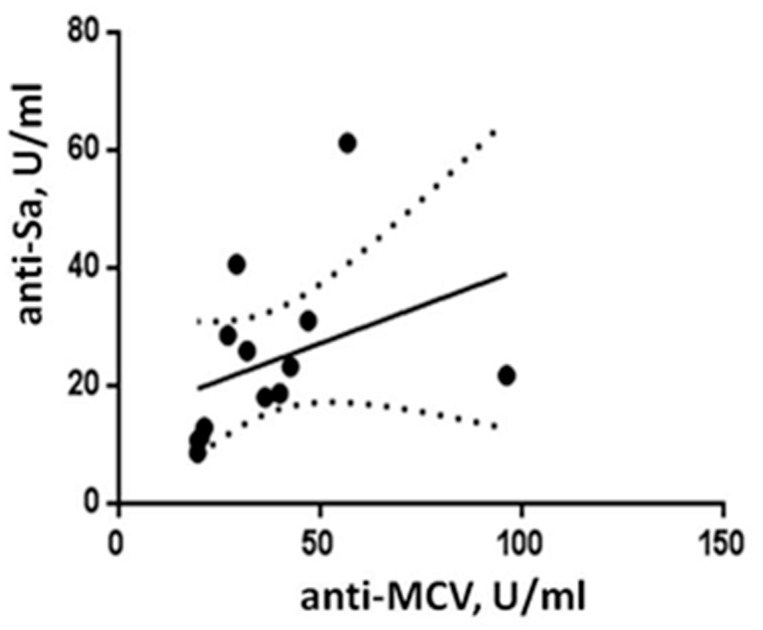
Correlation analysis of antibodies to mutated citrullinated vimentin (anti-MCV) and anti-citrullinated vimentin (anti-Sa) in patients with sarcoidosis.

**Figure 2 diagnostics-12-01939-f002:**
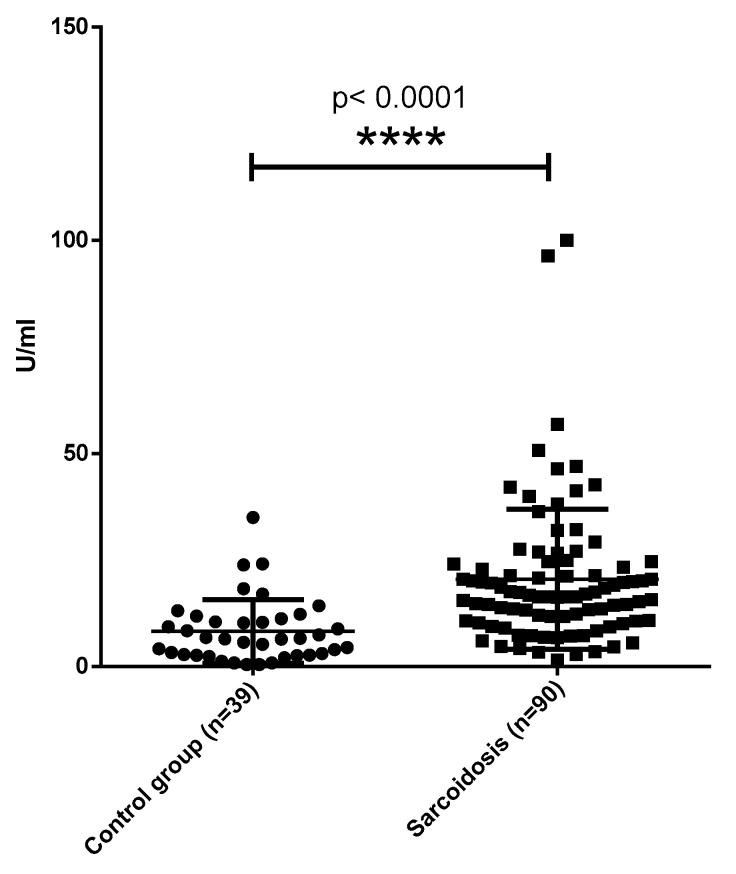
Levels of anti-vimentin autoantibodies in sera of patient with sarcoidosis and healthy subjects. **** *p* < 0.0001—in comparison with sarcoidosis and healthy subjects.

**Figure 3 diagnostics-12-01939-f003:**
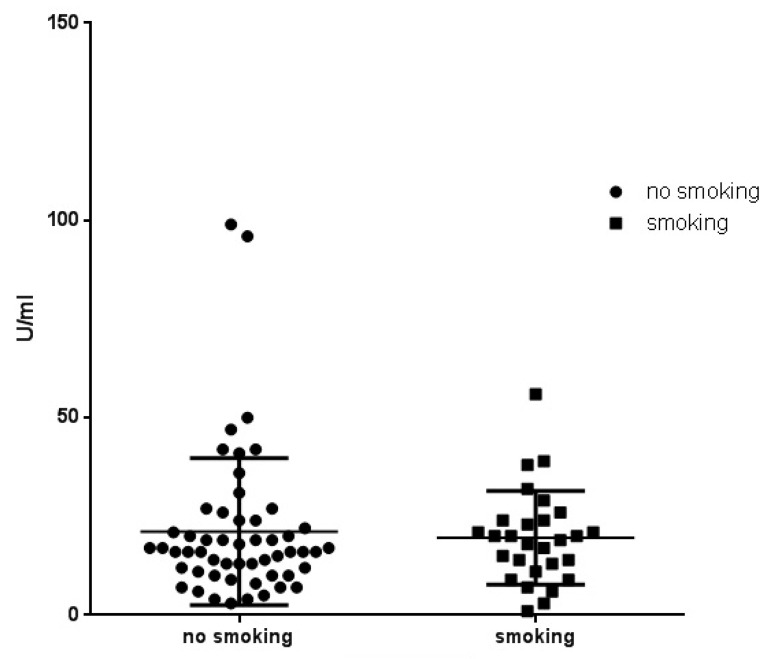
Antibodies to mutated citrullinated vimentin (anti-MCV) in smoking and non-smoking patients with sarcoidosis (*p* = 0.6180).

**Table 1 diagnostics-12-01939-t001:** Demographic characteristic of patients with sarcoidosis.

Characteristic of Patients	Sarcoidosis, *n* (%) (*n* = 93)
Gender	
Men	57 (61.1)
Women	26 (38.9)
Age	32.5 (±4.5) years
Morphological verification	93 (100.0)
Acute sarcoidosis (Löfgren’s syndrome)	12 (12.9)
Smoking	31 (33.3)
Allergy	16 (17.2)
Immunological tests for tuberculosis infection, number of positive results, *n* (%):	
T-SPOT.TB test	0/30
Mantoux test with 2 TU (>5 mm)	17/93 (18.3)
Chest CT-changes, *n* (%):	
Disseminative changes in the lungs	71 (76.3)
Intrathoracic lymphadenopathy	90 (96.7)
Infiltrative changes	13 (13.9)
Focus in the lungs	5 (5.3)
Complaints, *n* (%):	
General weakness	38 (40.8)
Fever	25 (26.9)
Cough	54 (58.1)
Dyspnea	27 (29.0)

**Table 2 diagnostics-12-01939-t002:** Anti-MCV and anti-CCP antibodies in patients with sarcoidosis and non-infectious lung diseases.

Studied Groups	Anti-MCV Results		CI 95%	Anti-CCP Results		CI 95%
	Positive Anti-MCV, %/(*n*/N)	Absolute Value (M ± m)		Positive Anti-CCP,	Absolute Value (M ± m)	
Sarcoidosis, *n* = 93	40.9 * (38/93)	20.31 ± 18.34	16.97–23.64	2.6 (1/38)	0.89 ± 0.39	1.17–2.64
Non-infectious lung diseases, *n* = 55	20.0 (11/44)	14.83 ± 15.58	10.53–19.12	15.4 2/13	2.43 ± 1.45	0.69–9.76
Healthy subjects (control group), *n* = 40	7.5 (3/40)	8.23 ± 7.44	5.82–10.65	0 (0/10)	0.55 ± 0.37	0.87–2.10

* *p* < 0.01—significant differences between sarcoidosis and control group.

**Table 3 diagnostics-12-01939-t003:** Anti-citrullinated vimentin (anti-Sa) levels in patients with sarcoidosis and non-infectious lung diseases.

Studied Groups	Patients with High Level of Anti-Sa (*n*, %)	Absolute Value (M ± m)	CI 95%
Sarcoidosis, *n* = 13	7/1353.8	0.89 ±13.09	14.87–27.02
Non-infectious lung diseases, *n* = 9	2/922.1	1.42 ± 71.89	4.67–17.98

**Table 4 diagnostics-12-01939-t004:** Antibodies to mutated citrullinated vimentin (anti-MCV) in patients with acute sarcoidosis (Löfgren’s syndrome).

Patients with Sarcoidosis, *n* = 93	Löfgren’s Syndrome Syndrome (*n* = 12)	Non- Löfgren’s Syndrome Syndrome (*n* = 81)	*p*-Level
Elevated levels of anti-MCV (*n*, %)	10 (83.3)	28 (34.6)	0.003

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
