# Peer review of "Detection of Anti-Vimentin Antibodies in Patients with Sarcoidosis"

_diagnostics, 2022, doi:10.3390/diagnostics12081939_

Round 1

Reviewer 1 Report

Dear Authors,

I have a few concerns about this manuscript.

1, Regarding to anti-vimentin antibodies determination, positive control was missing. Would be great if authors can provide standard curve. 

2, Please describe biological numbers and significance in Fig 1, Fig 2 and Fig 3. Graph quality was poor, authors should consider remake the graph. 

Author Response

  1. Regarding to anti-vimentin antibodies determination, positive control was missing. Would be great if authors can provide standard curve. 

Answer: The positive curve was constructed according to manufacturer's calibration samples which were provided in the kit. The cut-off was also chosen from the kit protocol. Therefore authors don't see the importance to include this data in graphs As it was mentioned in the methods section “For evaluation of positive autoantibody concentration the cutoffs proposed by the manufacturers were used”.

2, Please describe biological numbers and significance in Fig 1, Fig 2 and Fig 3. Graph quality was poor, authors should consider remake the graph. 

Answer: The information about statistical significance and the number of patients was provided in methods. We’ve also changed the Fig.2.

  1. The authors investigated the detection of antibodies in patients with sarcoidosis compared with patients with non-infection lung diseases and healthy subjects. Antibodies such as mcv and sa in patients with sarcoidosis were higher than the control groups, and the authors concluded that a high level of anti-MCV antibodies in patients with sarcoidosis characterizes the presence of an autoimmune process that is not related to citrullination. This conclusion includes an exaggeration. The least you could do is to describe an association with antibodies and sarcoidosis in Discussion.

Answer: In the conclusion section we pointed that “The low concentration of the anti-CCP antibodies and the positive correlation of anti-MCV and   anti-Sa antibodies suggest that citrullination and modification of vimentin is not a key factor in the development of an autoimmune response in patients with sarcoidosis”. The conclusion section in the abstract includes a mistake and was corrected.

Reviewer 2 Report

The authors investigated the detection of antibodies in patients with sarcoidosis compared with patients with non-infection lung diseases and healthy subjects. Antibodies such as mcv and sa in patients with sarcoidosis were higher than the control groups, and the authors concluded that a high level of anti-MCV antibodies in patients with sarcoidosis characterizes the presence of an autoimmune process that is not related to citrullination. This conclusion includes an exaggeration. The least you could do is to describe an association with antibodies and sarcoidosis in Discussion.

1.       Introduction: Please describe an association among MCV, CCp, and Sa.

2.       Table 2: What is the control group? Please describe the group in detail.

3.       Table 3: The authors mentioned that “anti-Sa antibodies were presented in the group of patients with sarcoidosis (n = 13) and in the subjects with non-specific lung diseases (n = 9). A high concentration of these antibodies was detected in 7 patients with sarcoidosis and in 2 patients with non-infection lung diseases.” These findings are different with results in Table 3. Moreover, how did you extract the 80 patients with non-infection lung diseases?

4.       Figure 1 shows the correlation of anti-sa and anti-mcv in 13 patients with sarcoidosis. However, block dots are twelve. Please recheck the data.

5.       Figure 2: Did you include the patients with non-infection lung diseases in control group? Together data in Table 2, the authors had better divide the control group into non-infection lung disease and healthy subjects.

6.       Discussion: The authors should describe limitations in this study.

Author Response

  1. Introduction: Please describe an association among MCV, CCp, and Sa

Answer:  Done.

  1. Table 2: What is the control group? Please describe the group in detail.

Answer: As it presented in the Materials section, “Disease control groups were: 55 with non-infectious lung diseases: 25 patients with chronic obstructive pulmonary disease (COPD), 15 patients with granulomatosis with polyangiitis and 15 with various alveolitis. The additional control group was comprised of 40 healthy volunteers with no chronic diseases, contacts with tuberculosis and changes in laboratory parameters”. The serum samples of the first control group were provided by the laboratory as a comparison group. All included patients were newly diagnosed according to the diagnostic criteria for each pathology and met the criteria for inclusion in this study. Detailed information is contained in the laboratory.

  1. Table 3: The authors mentioned that “anti-Sa antibodies were presented in the group of patients with sarcoidosis (n = 13) and in the subjects with non-specific lung diseases (n = 9). A high concentration of these antibodies was detected in 7 patients with sarcoidosis and in 2 patients with non-infection lung diseases.” These findings are different with results in Table 3. Moreover, how did you extract the 80 patients with non-infection lung diseases?

Answer: The table contained a typo error (corrected), the calculation was made for 9 patients with non-infection lung diseases

  1. Figure 1 shows the correlation of anti-sa and anti-mcv in 13 patients with sarcoidosis. However, block dots are twelve. Please recheck the data.

Answer: Probably there was a merger of two graph dots, because it was made based on 13 patients.

  1. Figure 2: Did you include the patients with non-infection lung diseases in control group? Together data in Table 2, the authors had better divide the control group into non-infection lung disease and healthy subjects.

Answer: Table 2 presents calculations separately for the main and two control groups. Figure 2 contains a comparison for the main group and the healthy control group (figure corrected)

  1. Discussion: The authors should describe limitations in this study.

Answer: Done

Reviewer 3 Report

The manuscript "Detection of Antibodies in Patients with Sarcoidosis" by Starshinova and colleagues examined the antibody response to vimentin on the risk of sarcoidosis. Chiefly, the authors investigated the presence of antibodies (anti-CCP and anti-Sa) to mutated citrullinated vimentin in serum in a cohort of 93 cases and 110 controls (i.e.., 55 non-infectious lung disease, 25 COPD, 15 granulomatosis and 15 alveolitis). Moreover, the authors also examined antibodies to non-modified vimentin in the same set of cases (n=93) against healthy controls (n=40). For all the studies, the authors used ELISA assay as a method of choice. The authors' findings showed high concentrations of anti-MCV antibodies in sarcoidosis patients compared to controls.

Thank you for allowing me to revise this. The work is of relevance in the field; however, I have some questions to be addressed in future revisions:

·       Regarding the epidemiological design, the authors included 93 sarcoidosis cases, 110 disease controls, and 40 healthy controls. However, it is not clear the 188 sera samples they referred to in the abstract section. Please address this issue. Also, in the material and methods, the authors stated a study population of 216 serum samples, of which 93 were sarcoidosis cases. Please clarify the epidemiological design for the study.

·       Within the 93 cases, the authors include 12 patients with Lofgren's syndrome (LS). It is known from previous immunological studies, as highlighted in reviews by Miedema et al. https://pubmed.ncbi.nlm.nih.gov/29310925/ and Kaiser et al. https://pubmed.ncbi.nlm.nih.gov/31000677/ that LS and non-Lofgren syndrome (non-LS) display different immunological profiles and have distinct disease mechanisms. Including LS patients may somewhat induce bias or dilate the results presented. To further characterize the antibody response, I suggest the authors include an additional analysis for LS and non-LS groups to distinguish sarcoidosis further.

·       Regarding including controls with other pulmonary diseases, please elaborate on whether the antibodies studied are also present in the disease controls. Please explain how these may impact the authors' findings. I also suggest including this as a limitation in the study.

·       Previous works suggest that smoking induces immune changes in vimentin (please see Bidkar et al. https://www.ncbi.nlm.nih.gov/pmc/articles/PMC5014446/) Klareskog et al. https://www.sciencedirect.com/science/article/pii/S1044532311000157. Therefore, the non-accountability of tobacco in the authors' analysis may bias the results. I suggest the authors add additional analysis to include smoking as a covariate in their investigation.

·       A limitations section of the study shall be included in the manuscript.

Author Response

The manuscript "Detection of Antibodies in Patients with Sarcoidosis" by Starshinova and colleagues examined the antibody response to vimentin on the risk of sarcoidosis. Chiefly, the authors investigated the presence of antibodies (anti-CCP and anti-Sa) to mutated citrullinated vimentin in serum in a cohort of 93 cases and 110 controls (i.e.., 55 non-infectious lung disease, 25 COPD, 15 granulomatosis and 15 alveolitis). Moreover, the authors also examined antibodies to non-modified vimentin in the same set of cases (n=93) against healthy controls (n=40). For all the studies, the authors used ELISA assay as a method of choice. The authors' findings showed high concentrations of anti-MCV antibodies in sarcoidosis patients compared to controls.

Thank you for allowing me to revise this. The work is of relevance in the field; however, I have some questions to be addressed in future revisions:

  • Regarding the epidemiological design, the authors included 93 sarcoidosis cases, 110 disease controls, and 40 healthy controls. However, it is not clear the 188 sera samples they referred to in the abstract section. Please address this issue. Also, in the material and methods, the authors stated a study population of 216 serum samples, of which 93 were sarcoidosis cases. Please clarify the epidemiological design for the study.

Answer: Corrected. There were 188 sera samples: 93 with sarcoidosis, 55 disease control (non-infection lung diseases) and 40 healthy controls

  • Within the 93 cases, the authors include 12 patients with Lofgren's syndrome (LS). It is known from previous immunological studies, as highlighted in reviews by Miedema et al. https://pubmed.ncbi.nlm.nih.gov/29310925/ and Kaiser et al. https://pubmed.ncbi.nlm.nih.gov/31000677/ that LS and non-Lofgren syndrome (non-LS) display different immunological profiles and have distinct disease mechanisms. Including LS patients may somewhat induce bias or dilate the results presented. To further characterize the antibody response, I suggest the authors include an additional analysis for LS and non-LS groups to distinguish sarcoidosis further.

Answer: In the Table 4 we demonstrated the results of aMCV in LS and non-LS groups and “Almost all patients with Lofgren’s syndrome included in the study had high levels of anti-MCV antibodies, which was statistically more frequent than in patients without Lofgren’s syndrome” and in further studies it will be usefull to distinguish these groups of patients.

  • Regarding including controls with other pulmonary diseases, please elaborate on whether the antibodies studied are also present in the disease controls. Please explain how these may impact the authors' findings. I also suggest including this as a limitation in the study.

Answer: Data on the detection of the studied antibodies in these groups are presented in the results section. As it was concluded, antibodies to citrullinated cyclic peptides are not significant in the pathogenesis of sarcoidosis  and other investigated pulmonary diseases (COPD, granulomatosis with polyangiitis, alveolitis).

  • Previous works suggest that smoking induces immune changes in vimentin (please see Bidkar et al. https://www.ncbi.nlm.nih.gov/pmc/articles/PMC5014446/) Klareskog et al. https://www.sciencedirect.com/science/article/pii/S1044532311000157. Therefore, the non-accountability of tobacco in the authors' analysis may bias the results. I suggest the authors add additional analysis to include smoking as a covariate in their investigation.

Answer: In the results section we “compared smokers and non-smokers among the patients with sarcoidosis. The influence of previous or present smoking to citrullination was not statistically significant (Fig.3)”. perhaps the sample size influenced this outcome and it is probably worth choosing groups more clearly taking into account smoking

  • A limitations section of the study shall be included in the manuscript –

Answer: Done

Round 2

Reviewer 1 Report

All my concerns are addressed. 

Reviewer 2 Report

The authors revised appropriately. No further correction is necessary.

Reviewer 3 Report

The revision has improved and the authors have addressed many of my comments.